



# Predicting extreme sub-hourly precipitation intensification based on temperature shifts

Francesco Marra[1,2,*], Marika Koukoula[3], Antonio Canale[4], and Nadav Peleg[3,*]

[1]Department of Geosciences, University of Padova, Padua, Italy
[2]Institute of Atmospheric Sciences and Climate, National Research Council, Bologna, Italy
[3]Institute of Earth Surface Dynamics, University of Lausanne, Lausanne, Switzerland
[4]Department of Statistical Sciences, University of Padova, Padua, Italy
[*]These authors contributed equally to this work.

**Correspondence:** Francesco Marra (francesco.marra@unipd.it), Nadav Peleg (nadav.peleg@unil.ch)

**Abstract.** Extreme sub-hourly precipitation, typically convective in nature, is capable of triggering natural disasters such as floods and debris flows. A key component of climate change adaptation and resilience is quantifying the likelihood that sub-hourly extreme precipitation will exceed historical levels in future climate scenarios. Despite this, current approaches to estimating future sub-hourly extreme precipitation return levels are deemed insufficient. The reason for this can be attributed to

two factors: there is limited availability of data from convective-permitting climate models (capable of simulating sub-hourly precipitation adequately), and the statistical methods we use to extrapolate extreme precipitation return levels do not capture the physics governing global warming. We present a novel physical-based statistical method for estimating the extreme sub-hourly precipitation return levels. The proposed model, named TEmperature-dependent Non-Asymptotic statistical model for eXtreme return levels (TENAX), is based on a parsimonious non-stationary and non-asymptotic theoretical framework that

incorporates temperature as a covariate in a physically consistent manner. We first explain the theory and present the TENAX model. Using data from several stations in Switzerland as a case study, we demonstrate the model's ability to reproduce sub-hourly precipitation return levels and some observed properties of extreme precipitation. We then illustrate how the model can be utilized to project changes in extreme sub-hourly precipitation in a future warmer climate only based on climate model projections of temperatures during wet days and on foreseen changes in precipitation frequency. We conclude by discussing the

uncertainties associated with the model, its limitations, and its advantages. With the TENAX model, one can project sub-hourly precipitation extremes at different return levels based on daily-scale projections from climate models in any location globally where observations of sub-hourly precipitation data and near-surface air temperature are available.

## 1 Introduction

Extreme sub-hourly precipitation can lead to natural disasters such as flash floods, urban floods, and debris flows (Borga et al.,

2014; Cristiano et al., 2017). Quantifying the probability of exceedance of sub-hourly extreme precipitation in future climate scenarios is thus of high interest for climate change adaptation and resilience (Westra et al., 2014; Fowler et al., 2021b). According to thermodynamics, the atmospheric water vapor holding capacity increases with temperature at an exponential rate





($\sim 7\%\,°C^{-1}$, Clausius-Clapeyron relation; Trenberth et al., 2003). In the presence of full saturation and maximum precipitation efficiency, conditions that are closely met during sub-hourly extreme events, precipitation intensities are expected to increase

with temperature at a similar rate; this has been demonstrated at the global scale (e.g., Ali et al., 2021). Evidence shows that the observed scaling rates may significantly deviate from the Clausius-Clapeyron relation, sometimes by up to threefold (Pfahl et al., 2017; Fowler et al., 2021a). This is likely caused by temperature-induced changes in the local atmospheric dynamics (e.g., vertical advection, moisture convergence) and strongly depends on the temporal (Fowler et al., 2021a) and spatial (Peleg et al., 2018) scales of interest. Climate change is thus expected to modify precipitation extremes in complex ways, due to the

interplay of local dynamic and thermodynamic processes and large-scale atmospheric dynamics.

Precipitation extremes that occur very rarely, such as magnitudes that are exceeded with a low probability $p$ in a given year (usually referred to as $\mathcal{T}$-year return levels, with $\mathcal{T} = 1/p$), are critical for the design and management of risk mitigation plans. For example, urban drainage systems in many countries are designed to cope with rainfall intensities up to a certain $\mathcal{T}$-year return level, beyond which urban flooding is expected. Therefore, it is essential to estimate future precipitation return levels

to facilitate climate change adaptation. These extremes cannot be derived directly from the available observations and need to be extrapolated. To do so, hydrologists and practitioners often apply extreme value analysis methods, which are typically cumulative distribution functions of the annual maximum precipitation intensities (e.g., Papalexiou and Koutsoyiannis, 2013). These models are usually described by parameters that are estimated from the available extremes, either the annual maxima or the exceedances of a high threshold, assuming stationarity (Coles, 2001; Kats et al., 2002). Climate change, however,

undermines this stationarity assumption.

Quantifying the impact of climate change on extreme sub-hourly precipitation is typically accomplished in one of three ways. One option is to compute the relative change in precipitation return levels for different time intervals (e.g., present and future) obtained directly from climate model simulations while assuming that stationarity holds in each time interval (for example see Ban et al., 2020; Moustakis et al., 2021). A second option is to use observed precipitation data to bias-correct

precipitation time series obtained from climate models, computing and examining the changes in precipitation extremes over present and future periods (e.g., Maity and Maity; 2022, Yan et al., 2021). Sub-hourly data from climate models, however, are rarely available due to storage limitations. In addition, climate models at the global and regional scales cannot resolve processes that are critical for extreme sub-daily precipitation, such as convection, making most climate models unsuitable for analyzing extreme sub-daily precipitation. Convection-permitting models can be used for such a task (Ban et al., 2020) but as

they require high computational resources the available simulations only cover a few regions, are forced by few of the socio-economic pathways, and are made available with delay with respect to state-of-the-art global climate models (e.g., Fosser et al., 2020).

The last option is to utilize non-stationary extreme value models whose parameters depend on a covariate, such as time or temperature (e.g., Cheng and AghaKouchak, 2014; Sippel et al., 2015; Pfahl et al., 2017; Vidrio-Sahagún and He, 2022),

and to extrapolate the information to future scenarios based on this covariate. As a general rule, some of the parameters of the extreme value distribution are kept unchanged or linked to one another to reduce uncertainties in parameter estimation (Prosdocimi and Kjeldsen, 2021). The dependence of the other parameter(s) on the covariate of interest is chosen among a





set of suitable models (Ragno et al., 2019). Traditional extreme value distributions, however, cannot be easily linked to the underlying physical processes because their parametrization does not allow to separate the contribution of thermodynamics

and atmospheric dynamics (Marra et al., 2021). The dependence of extremes on the covariate of interest is thus empirical and often limited to monotonic relations. Extrapolating beyond the training period becomes highly uncertain (Serinaldi and Kilsby, 2015; Fatichi et al., 2016; Iliopoulou and Koutsoyiannis, 2020; Tabari, 2021).

Temperature can be considered a primary candidate for a covariate in non-stationary extreme value models as a result of its direct physical relation with extreme precipitation described above. Several studies have found that extreme precipitation

(defined as precipitation corresponding to the 95th or 99th percentile of the wet time intervals) increases exponentially (i.e., linearly on a logarithmic scale) as near-surface air- or dew-point temperature increases. This behavior is often referred to as extreme precipitation–temperature scaling (for a review of the topic, see Westra et al., 2014; and Fowler et al., 2021c). This relation is derived for the entire sub-daily precipitation and temperature data available, often using either a binning method (e.g., Ali et al., 2021) or a quantile regression method (Wasko and Sharma, 2014). Under some assumptions, it is possible to project

changes in precipitation extremes using these approaches (Peleg et al., 2022). This, however, comes with several limitations. First, the projection will be limited to the percentile under consideration and $\mathcal{T}$-year return levels cannot be extrapolated as they correspond to percentiles that are too high to be derived empirically. Second, return levels also depend on the occurrence frequency of precipitation and the current methods for predicting changes in precipitation extremes with temperature have the disadvantage of assuming that the storm frequencies will remain unchanged, which is a strong and unlikely assumption. Last,

in many locations, we observe a break in the extreme precipitation-temperature exponential relationship at high temperatures (Drobinski et al., 2016), which poses doubts about the accuracy of the extrapolation (Yin et al., 2021). The reasons for this break, also known as "hook structure", are twofold: limitation in the humidity supply at high temperatures that prevent precipitation initiation, and insufficient precipitation data at high temperatures due to the rarity of very hot and wet conditions. Current methods for predicting extreme precipitation intensification do not take this factor into account, which is a significant

drawback. Using current methods it is thus impossible to extrapolate how sub-hourly precipitation at a specific $\mathcal{T}$-year return level will intensify as the temperature increases.

In light of this, we argue that current approaches to estimating future precipitation return levels cannot adequately capture the physics governing climate change. Sub-hourly extremes cannot yet be quantified in a physically consistent manner. Here, we present a new method to derive projections of extreme sub-hourly precipitation return levels based on in-situ observations

of precipitation and temperature, on climate model projections of temperatures during wet days, and on the projected changes in the frequency of precipitation events. The model is based on a parsimonious non-stationary and non-asymptotic statistical framework that uses temperature as a covariate in a physically consistent manner.

## 2   The TENAX model

The idea behind the proposed model is to separate the physical dependence of extreme precipitation on temperature from the

occurrence of precipitation events at a given temperature. For this purpose, we combine (Fig. 1): (i) a non-stationary statistical





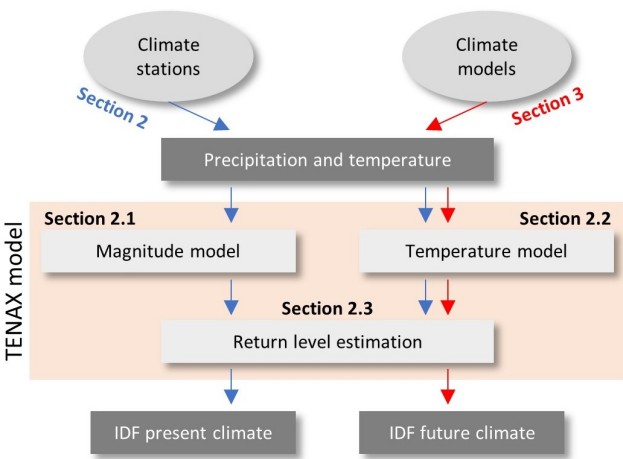

**Figure 1.** A schematic illustration of the components, data, and outputs of the TENAX model.

model for the cumulative distribution function of the precipitation event magnitudes that uses temperature as a covariate; and (ii) an analytical probability density function for temperatures during precipitation events. Combining these models with (iii) a non-asymptotic formulation for extreme return levels, we derive the TEmperature-dependent Non-Asymptotic statistical model for eXtreme return levels (TENAX). Non-asymptotic statistics rely on the idea that extremes are samples from the set

of independent realizations of the process of interest, which are usually termed "ordinary events" or simply "events". These methods allow one to write extreme value distributions based on the cumulative distribution function of the ordinary events and on their occurrence frequency (see Marani and Ignaccolo, 2015; Marra et al., 2019), exploiting the fact that the cumulative distribution function $G(x)$ of the precipitation annual maxima emerging from a finite number $n$ of independent events per year sampled from the cumulative distribution function $F(x)$ can be written as

$$G(x) = F(x)^n. \tag{1}$$

Denoting with $W(x;T)$ the cumulative distribution function of the magnitude of the ordinary events occurring at a temperature $T$ and with $g(T)$ the probability density function of temperatures at which the precipitation events occur, the (marginal) parent cumulative distribution function of the events magnitudes $F(x)$ becomes:

$$F(x) = \int_{-\infty}^{+\infty} W(x;T) \cdot g(T) \, dT. \tag{2}$$

In Section 2.1 we present the precipitation event magnitude model $W(x;T)$, followed in Section 2.2 by the temperature model of the events $g(T)$. As the analytical expression for $F(x)$ may be difficult to treat, we also present in Section 2.3 an alternative to derive the cumulative distribution function of the emerging annual maxima using a Montecarlo procedure. In this way, only the analytical expressions for the two model components are needed.





## 2.1 Precipitation event magnitude model

We define independent ordinary precipitation events using the unified framework proposed by Marra et al. (2020), which proceeds in two steps: (i) independent "storms" are defined as wet periods separated by dry intermissions of at least $d_{\mathrm{dry}} = 24$ hours; and (ii) ordinary events of duration $d$ are defined as the maximum $d$-duration intensity observed during each storm. For this purpose, we use a running window with size $d$ and time steps equal to the temporal resolution of the data. It was shown that using this framework the ordinary events share the statistical properties of the $d$-duration annual maxima (e.g., the scaling

with duration) for all durations $d \leq d_{\mathrm{dry}}$ (Marra et al., 2020).

We use the Weibull distribution to model the magnitudes of sub-hourly ordinary precipitation events (referred to as "events" hereafter). This model, with powered-exponential tails, is justified by thermodynamic arguments (Wilson and Toumi, 2005) and is supported by empirical evidence, especially in the case of convective precipitation, which is the main driver of hourly and sub-hourly extremes (Berg et al., 2013; Marra et al., 2020; Wang et al., 2020; Dallan et al., 2022; Marra et al., 2022;

Papalexiou, 2022). The Weibull tail model describes the non-exceedance probability of magnitudes higher than a defined left-censoring threshold ($\vartheta^*$) using two parameters (scale and shape). These parameters can be explicitly dependent on a covariate, such as the near-surface air temperature $T$:

$$W(x;T) = 1 - \mathrm{e}^{-\left[\frac{x}{\lambda(T)}\right]^{\kappa(T)}}, \tag{3}$$

where $\lambda$ and $\kappa$ are the scale and shape parameters, respectively. It is important to note that, although we use near-surface air

temperature as $T$ in our formulation and presentation of the model, dew-point temperature can also be used as an alternative covariate. As a matter of fact, some consider it to be a superior choice (e.g., Wasko et al., 2018; Ali et al., 2021). Based on our analyses of the case study presented later, we found no appreciable difference between near-surface temperature and dew-point temperature (not shown).

The Clausius-Clapeyron relation suggests an exponential dependence of extreme precipitation with temperature, as con-

130 firmed by multiple studies (as an example, see Fowler et al., 2021c). This translates into an exponential dependence of the scale parameter $\lambda$ on temperature $T$:

$$\lambda(T) = \lambda_0 \cdot \mathrm{e}^{aT}. \tag{4}$$

Since the scaling of extreme precipitation with temperature sometimes also depends on the quantile examined (Lenderink and van Meijgaard, 2008; Hardwick Jones et al., 2010), it follows that the shape parameter $\kappa$ may depend on temperature as

well. This dependence is not obvious and may be masked by the uncertainty characterizing the estimation of this parameter (as later discussed). Here we propose a simple linear relation:

$$\kappa(t) = \kappa_0 + bT. \tag{5}$$

The parameters of the magnitude model are estimated by left-censoring the observations below a properly defined threshold $\vartheta^*$ and using the Maximum Likelihood method. The left-censoring threshold depends on the local climate and can be identified





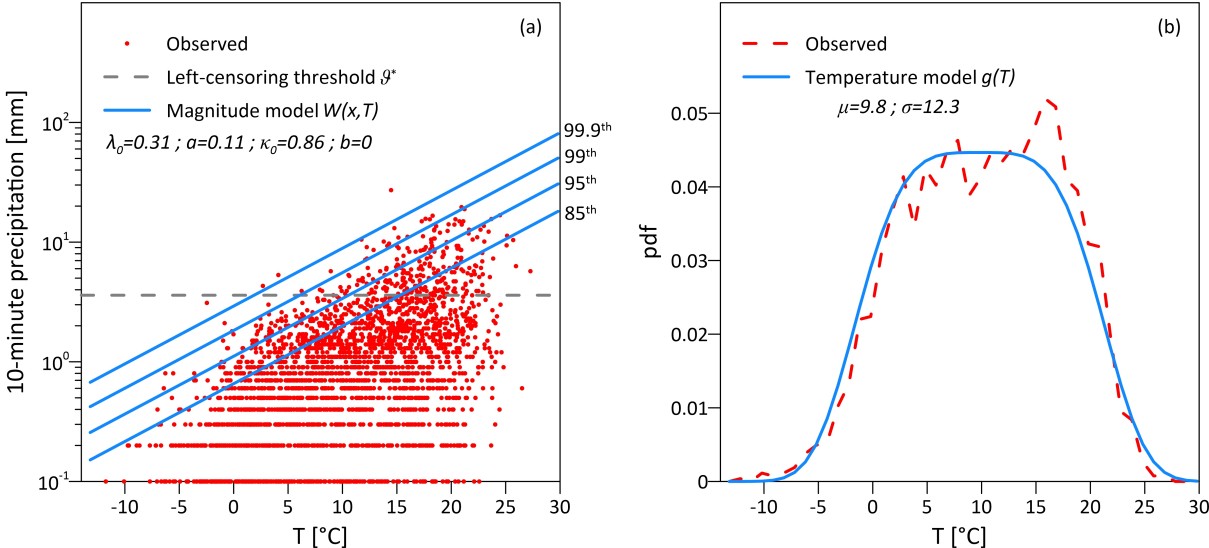

**Figure 2.** The two model components for the Aadorf station. (a) Empirical observations (red dots) and magnitude model (blue lines corresponding to different percentiles); the dashed grey line shows the left-censoring threshold used. The parameter $b$ is not significantly different from zero, only three parameters $(\lambda_0, a, \kappa_0)$ are used. (b) Empirical probability density of the average temperatures observed during the 24 hours preceding the 10-minute peak precipitation intensities (dashed red) and the estimated Generalized Gaussian temperature model $g(T)$ (solid blue).

using objective tests (Marra et al., 2023). Here we use a generic threshold of $\vartheta^* = 0.9$ (i.e., the local 90th percentile of the ordinary events). This threshold was found adequate for sub-hourly precipitation extremes in a variety of climatic conditions (e.g., Wang et al., 2020; Marra et al., 2020; Dallan et al., 2022; Marra et al., 2022). Note that this threshold is not a parameter that needs to be optimized for each case. In fact, provided that Weibull is an adequate tail model, any threshold $\vartheta \geq \vartheta^*$ provides indistinguishable results (Marra et al., 2019). This implies that the sensitivity of the results on this parameter, once it is properly 145 defined, is quite low. The magnitude model has four parameters $(\lambda_0, a, \kappa_0, b)$. The statistical significance of the dependence of the shape parameter on $T$ (i.e., the significance of the $b$ parameter being different from zero) can be evaluated using the Likelihood-ratio test.

The example to follow focuses on 10-minute peak precipitation intensities ($d = 10$ min) as a proxy of generic sub-hourly intensities. Our analyses show that the model assumptions generally hold for durations between 10 minutes and 1 hour, and 150 even longer. We fitted the magnitude model (Fig. 2a) by using data from the Aadorf station in Switzerland (see the description of the station and data in Section 4). In this case, the parameter $b$ was found to be not significantly different from zero at level 5%. Therefore, we set $b = 0$ and estimated the three remaining parameters of the magnitude model $(\lambda_0, a, \kappa_0)$ accordingly.



## 2.2 Temperature model

We find that in our study case the average temperatures observed during $D$ hours preceding the peak intensities are well
described by a Generalized Gaussian distribution with shape parameter 4 (e.g., Fig. 2b), whose probability density function is:

$$g(T) = \frac{2}{\sigma \cdot \Gamma(1/4)} \cdot \exp\left[-\left(\frac{T-\mu}{\sigma}\right)^4\right], \tag{6}$$

where $\mu$ and $\sigma$ are location and scale parameters, respectively. The parameters $\mu$ and $\sigma$ can be estimated using the Maximum
Likelihood method.

We explored time intervals $D$ ranging from 1 hour to 24 hours in our case study without observing significant deviations
from this Generalised Gaussian model (not shown). We focus here on the case $D = 24$ hours, as daily temperatures are easier
to derive from climate model simulations and are therefore preferred for climate change projections.

It is interesting to note that the Generalized Gaussian model can emerge from the combination of two normal distributions
with similar variance and different means, and can thus be interpreted as the coexistence of different precipitation types that
occur at different mean temperatures (e.g., summer/winter precipitation, or stratiform/convective processes – Molnar et al.,
2015). For the Aadorf station, for example, separating summer (May to October) and winter (November to April) events yields
two normal distributions with similar variance and different mean (Fig. 3). Super-positioning the two normal distributions as a
two-component mixture model is well approximated by the Generalized Gaussian model (Fig. 3). The Generalized Gaussian
fit outperforms ordinary Gaussian fit to the data (Fig. S1). This is especially true for the right tail of the temperature model that,
due to the positive scaling of precipitation intensities with temperature, influences the distribution of extremes the most (e.g.,
see Fig. S2 where the model was fitted with both options). In other study areas, different approximations of this combination
or even of different temperature models could be required. Future research should investigate in greater detail the temperature
model in different regions as well as the underlying drivers.

## 2.3 Return level estimation

The cumulative distribution function $F(x)$ defined in Eq. 2 has a complex analytical form. However, once the magnitude
model $W(x; T)$ and the temperature model $g(T)$ are defined, the temperature-dependent non-asymptotic distribution for the
extreme return levels (TENAX) can be derived using a Montecarlo framework. Specifically, it is possible to generate a large
collection of temperatures $T_i$ with $i = 1, \ldots, N$ sampled from $g(T)$ to obtain a Montecarlo approximation of $F(x)$. Using the
Simplified Metastatistical Extreme Value formulation (SMEV) formulation (Marra et al., 2019), we can obtain an estimate of
the distribution of annual maxima as:

$$G_{\text{TENAX}}(x) = \left(\int\limits_{-\infty}^{+\infty} W(x;T) \cdot g(T)\, dT\right)^n \simeq \left(\frac{1}{N}\sum_{i=1}^{N} W(x;T_i)\right)^n, \tag{7}$$

where $N$ is the number of Montecarlo-generated events and $n$ is the average number of events in a year. Return levels can be
derived numerically by inverting Eq. 7. In our case study, we set $N$ to $2 \cdot 10^4$.





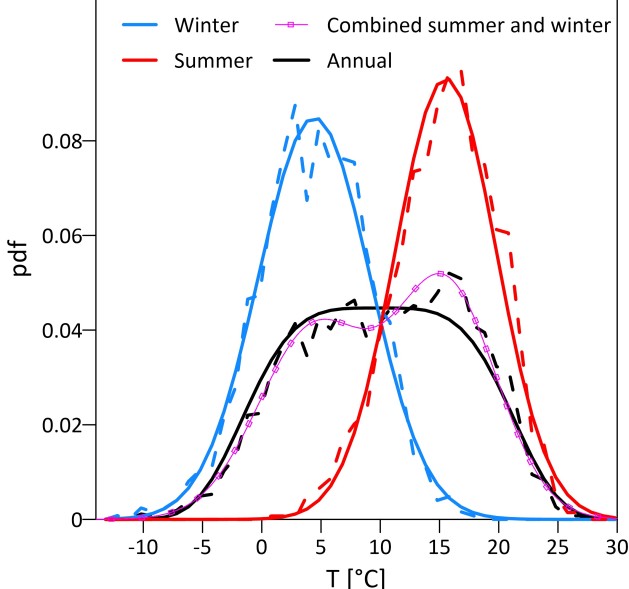

**Figure 3.** Empirical probability densities of the average temperatures observed during the 24 hours preceding the 10-minute peak precipitation intensities (dashed lines) for precipitation events during the entire year (black; as in Fig. 2), summer events (red, May to October), and winter events (blue, November to April). Solid lines represent the fitted temperature models for the entire year (black, the $g(T)$ model; as in Fig. 2), summer (red, Gaussian model), and winter (blue, Gaussian model). The purple line shows the combination of the empirical summer and winter distributions.

It is important to point out that the tail heaviness of $F(x)$ in Eq. 2 depends on the properties of both the magnitude model $W(x;T)$ and the temperature model $g(T)$. Therefore, the independence of the shape parameter of the magnitude model on temperature (i.e., the case $b = 0$) does not imply that the tail heaviness of $F(x)$ is invariant (see also Fig. S2). Moreover, despite the magnitude model $W(x;T)$ is based on a Weibull distribution, combining it with temperature models with heavier tails than the one in Eq. 6 (e.g., a Gaussian model; Fig. S1) yields parent distributions with tails that are heavier than Weibull (Fig. S2). Thus, the TENAX model with Weibull tails can explain the tails heavier than Weibull reported in some cold regions, e.g., Northern Europe (Wang et al., 2020; Poschlod, 2021). This showcases the flexibility of the TENAX model formulation.

## 2.4 Uncertainty quantification

The estimated precipitation return levels are subject to uncertainties arising both from the magnitude and temperature models. In principle, it is possible to quantify the asymptotic variance of the parameters' estimators under the maximum likelihood framework, but it would then be complex to propagate those to the return levels. It is therefore more convenient to use a bootstrap approach, such as the one suggested by Overeem et al. (2008). The years in the record are randomly re-sampled with replacement to create multiple realizations of $M$ years each, where $M$ is the observed record length. The model parameters and precipitation return levels are then estimated for each realization to obtain an estimate of the uncertainty. This method, already





in use for other non-asymptotic approaches (e.g., Marra et al., 2020), ensures that the influence of long-term drivers that act on seasonal or annual levels (i.e., the natural climate variability) is preserved. This allows us to explicitly consider uncertainty in both the estimation of the average annual number of events and the temperature affecting the events.

## 2.5    Validation of the TENAX model

The TENAX model is validated in light of its performance in quantifying extreme precipitation at different return levels, as well as its ability to reproduce the properties of the extreme precipitation-temperature scaling relationship. We first compare the return levels estimated using the TENAX model with (i) the official return levels provided by MeteoSwiss for the Aadorf station; and (ii) the return levels estimated using an established non-asymptotic method (Fig. 4). As can be seen in the figure,
the combination of the magnitude model $W(x;T)$ and temperature model $g(T)$ in the Montecarlo framework of Eq. 7 provides return level estimates that are indistinguishable from those of both traditional methods (GEV-based estimation by MeteoSwiss) and established non-asymptotic approaches (the SMEV model with Weibull parent distribution and the same left-censoring threshold as the TENAX model; Marra et al., 2020). For example, MeteoSwiss and SMEV estimations for the 10-year return levels (50-year) are 19.1 mm and 19.8 mm (26.4 mm and 28.9 mm), respectively, while the TENAX estimations are in close
agreement with 20 mm (29.7 mm) prediction (Fig. 4). In addition, uncertainties in the estimated return levels are comparable to the ones of the SMEV model and are smaller than the ones of the official MeteoSwiss estimates despite being based on data records that are over 20 years shorter (1981-2018 of SMEV and TENAX as opposed to 1960-2020 of MeteoSwiss).

To validate the TENAX model's ability to reproduce the extreme precipitation-temperature scaling relationship we computed the 99th percentile of the 10 min peak precipitation intensities from the observed data using both the quantile regression
approach suggested by Wasko and Sharma (2014) and the common temperature binning approach (e.g., Ali et al., 2018) with fixed intervals of 1.5°C (Fig. 5). The 99th scaling rate using the quantile regression method was found to be $12.3 \pm 0.8 \%\,°\text{C}^{-1}$; we note that this rate is higher than the expected rate for this region (around $7\%\,°\text{C}^{-1}$; Molnar et al., 2015) since the scaling was computed for the event peak intensities rather than for all the 10-min wet time intervals. Plotting the magnitude model $W(x;T)$, we obtained a visually good fit with the observed scaling (Fig. 5), as well as an agreement with the scaling rate, which
in the TENAX model is linked to the parameter $a$ ($11.8\%\,°\text{C}^{-1}$). The TENAX model not only fully matches the precipitation-temperature scaling right (Fig. 5), but also reproduces correctly the break in scaling relationship, known as "hook structure" and reported by many studies (e.g., Utsumi et al. 2011; Panthou et al. 2014; Visser et al., 2021; Yin et al. 2021). This is an important result, as TENAX was not explicitly designed to reproduce this phenomenon. The "hook structure" naturally emerges from an exponential dependence of extreme precipitation on temperature that has no intrinsic upper limit (as Clausius-Clapeyron)
and from a sharp decrease in the probability of occurrence of precipitation events at high temperatures. For Aadorf, this sharp decrease appears to be well reproduced by a Generalized Gaussian model with a shape parameter equal to 4, as in Eq. 6.





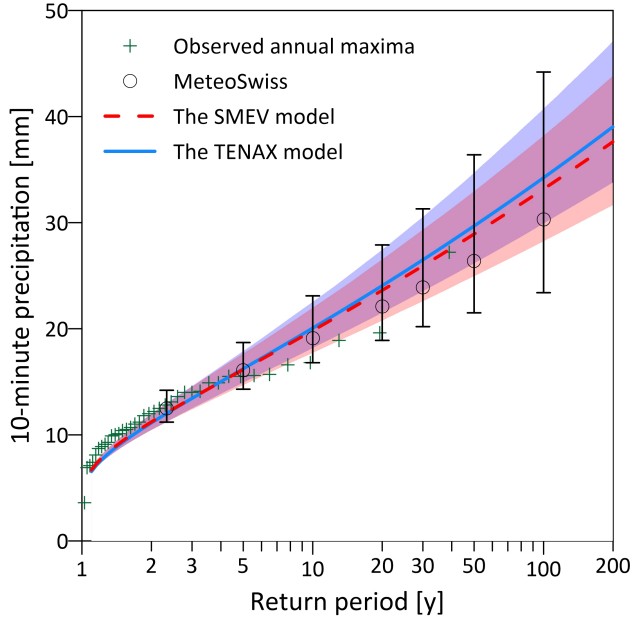

**Figure 4.** Observed annual maxima for the station of Aadorf plotted using the Weibull plotting positions (green crosses) and (i) MeteoSwiss official return levels for a set of return periods (black circles; error bars show the 5-95th confidence interval); (ii) return levels estimated using a one-type SMEV (red dashed line; shaded red area shows the 5-95th confidence interval obtained from $10^3$ bootstraps with replacement across the available years); (iii) return levels estimated using the TENAX model ($G_{\text{TENAX}}(x)$ in Eq. 7, blue solid line; shaded blue area shows the 5-95th confidence interval obtained from $10^3$ bootstraps with replacement across the available years).

## 3 Climate change projections

The magnitude model $W(x;T)$ represents the physics of the precipitation processes at a given temperature in the area of interest. Assuming that this physics is invariant (for example, that the scaling relationship between extreme precipitation and temperature is maintained, as shown by Ban et al., 2020), it is possible to use the TENAX model to derive projections of future return levels based on (i) the projected changes in mean and variance of the temperatures affecting the precipitation events and on (ii) the projected changes in average number of annual precipitation events. To do so, one needs to derive a projected temperature distribution $g'(T)$ and apply the Montecarlo method in Eq. 7 using the projected number of annual events $n'$.

The temperature distribution $g'(T)$ can be derived based on the projected changes in mean and standard deviation of the $D$-hour temperatures preceding the peak precipitation intensities. We found that the 24-hour temperatures preceding the peak precipitation intensities can be approximated by changes in the daily temperatures during precipitation events. Hence, the advantage of using $D = 24$ hours and not a shorter duration to estimate the change in temperature becomes now clear, as it is possible to use climate models with daily temperature and precipitation data for this purpose. For example, daily simulations are available from numerous regional climate models and socio-economic pathways, as opposed to convection-permitting simulations that are currently available for only a limited number of emission scenarios. In principle, to adjust accurately the





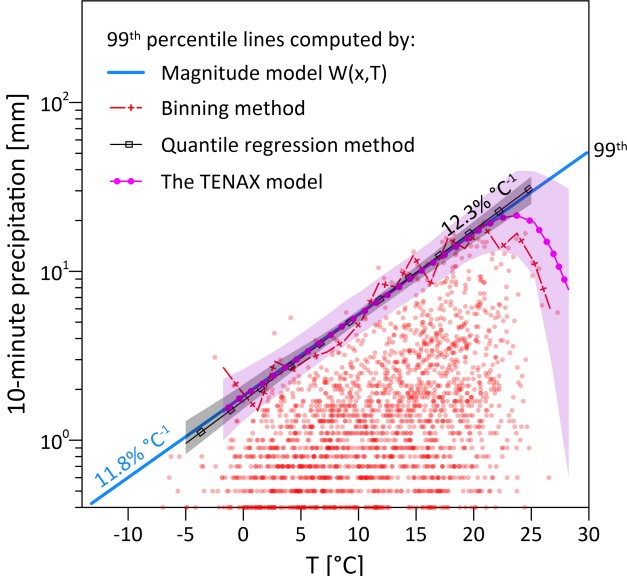

**Figure 5.** Precipitation-temperature scaling relation for the station of Aadorf. The magnitude model $W(x;T)$ (blue line) reproduces the observed extreme precipitation-temperature scaling rate for the 99th percentile as obtained from a quantile regression method (black squared-line; shaded area shows the 5-95th confidence interval obtained from $10^3$ bootstraps with replacement across the available data points) and from using the binning method (red dotted line). The 99th percentile scaling line simulated by the TENAX model is plotted with a purple dotted line (median from $10^3$ Montecarlo samples with the same number of events of the observed record; the shaded area shows the 5-95th confidence interval). The numerical values represent the 99th scaling rate for the observed quantile regression method (black) and the modeled magnitude model $W(x;T)$ (blue).

average number of precipitation events in the future, climate models with at least hourly temporal resolution are required. However, since the sensitivity of the return levels to this parameter is relatively low (see below), one could use the changes in the number of wet days as a proxy for the change in the number of precipitation events.

This approach to project future precipitation return levels relies not only on the assumption that the magnitude model $W(x;T)$ is invariant but also that it holds for unexplored higher temperature ranges. While it is safe to assume that the dependence on temperature of a given physical process remains invariant (Trenberth et al., 2003), it is necessary for the entire $W(x;T)$ to be invariant. This is a stronger assumption as it also requires the proportions among different processes leading to heavy precipitation to be unchanged. We claim this assumption is reasonable for the case of sub-hourly precipitation, as sub-hourly extremes tend to be related to convective processes only. Additional care should be taken when applying similar

models to longer durations for which multiple processes characterized by different magnitudes may lead to extremes, such as daily durations. Here, extensions of the TENAX model able to handle multiple types of processes (e.g., Marra et al., 2021) may be required.




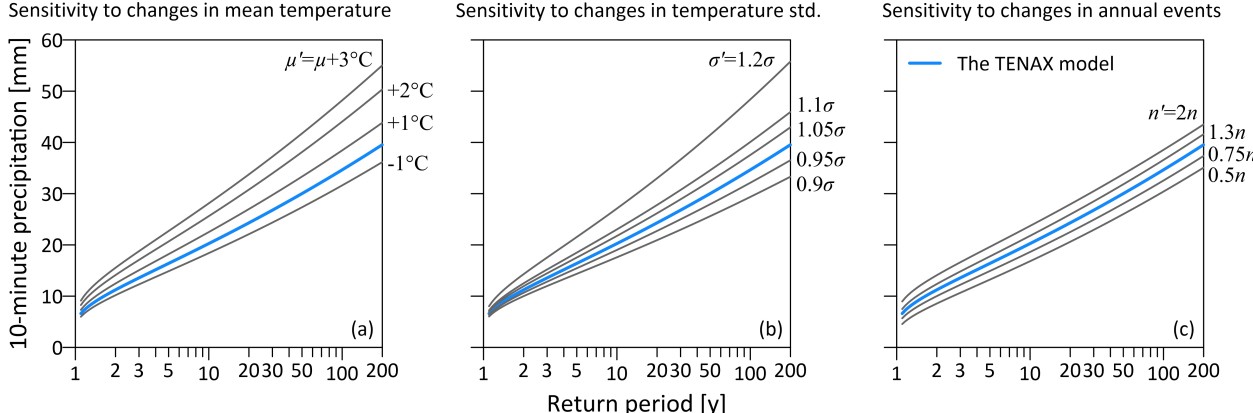

**Figure 6.** Sensitivity of sub-hourly precipitation return levels of the Aadorf station to changes in the temperature model (mean temperature $\mu$, panel a, and temperature standard deviation $\sigma$, panel b) and in the number of annual precipitation events ($n$, panel c).

## 3.1 Sensitivity of the projections to $\mu, \sigma$, and $n$

Next, we examined the sensitivity of the modeled extreme sub-hourly precipitation return levels to changes in climate. Figure
6 shows the sensitivity to changes in (i) the mean temperature during a $D$-hour interval preceding the 10-minute peak precipitation intensity $\mu$ (Fig. 6a); (ii) the corresponding standard deviation $\sigma$ (Fig. 6b); and (iii) the average yearly number of precipitation events $n$ (Fig. 6c).

Increases in mean temperature $\mu$ and/or in the standard deviation $\sigma$ imply a higher probability of occurrence of precipitation events at higher temperatures, which through the magnitude model is related to a higher probability of precipitation extremes.
Interestingly, $20\%$ changes in the standard deviation yield a similar impact on precipitation return levels as of $3°$C increase in the mean temperature. The mean change in temperature during precipitation alone could then be insufficient to fully describe the expected changes in extremes, with critical implications for climate change projections.

The sensitivity of extreme precipitation return levels to $n$ is relatively small. The average number of precipitation events in a year needs to double (i.e., $2n$) to have an impact comparable to a $1°$C increase in the mean temperature or to a $\sim 10\%$ increase
in the standard deviation (i.e., $1.1\sigma$). The other stations examined in the case study also demonstrate that temperature changes have a greater impact on precipitation return levels than changes in the number of precipitation events (not shown).

## 3.2 Validation of the projections

We validated the ability of the TENAX model to project precipitation return levels under increased temperatures in hindcast, by splitting the 38-year record of the Aadorf station into two periods of 19 years each. The mean temperature during precipitation
events $\mu$ increased by $0.46°$C for the second period (2000-2018) compared to the first period (1981-1999), accompanied by an increase of $6\%$ in the standard deviation of temperature during precipitation events $\sigma$ and by a $1.8\%$ increase in the average





number of events per year $n$. This is confirmed by the different empirical probability distributions of temperature during the events between the two periods (Fig. 7a, dashed lines).

A magnitude model $W(x;T)$ was fitted independently for each period, and their similarity was checked. The null hypothesis
of the magnitude models being the same in the two periods could not be rejected (likelihood ratio test), thus supporting our working hypothesis; i.e., that a magnitude model fitted for an observed period will be invariant and can be used in a future-warmer period. Notably, the similarity between magnitude models of the two periods holds for almost all the stations presented later in the case study (not shown), implying that changes in extreme sub-hourly precipitation are likely entirely and directly driven by changes in temperature. The only exception (Adelboden station) is due to an individual outlier (a particularly strong
winter precipitation event occurred at cold temperature) that heavily affects the magnitude model in one of the two periods.

While the temperature differences may seem small, they imply a considerable increase in the 10-minute peak precipitation return levels, as demonstrated by the fitted SMEV models to both periods (dashed lines in Fig. 7b), which are in agreement with the observed annual maxima (plus symbols in Fig. 7b). We then implemented the TENAX model, to predict the precipitation return levels for the second period. To that end, we used the magnitude and temperature models fitted for the first period
but shifted the $\mu$ and $\sigma$ parameters of the temperature model to fit the temperature of the second period (i.e., by applying $\mu' = \mu + 0.46$; $\sigma' = 1.06 \cdot \sigma$; $n' = 1.018 \cdot n$). As a result, we obtained the precipitation return levels shown with a solid red line in Fig. 7b. The results obtained from the SMEV and TENAX models for the second period (Fig. 7b; dashed and solid red lines, respectively) are very similar, although no information on the magnitude of any event that occurred during the second period is used for this projection. The numerical experiment demonstrated that the TENAX model can be applied effectively to study
the effects of climate change on precipitation return levels.

Beyond examining the changes in the observed annual maxima between the two periods (plus symbols in Fig. 7b), other indications support our model results of the intensification of short-duration precipitation extreme. For example, Libertino et al. (2019) reported a statistically significant increase in short-duration (hourly) precipitation return levels in the Italian Alps, not far from the Aadorf station. Dallan et al. (2022) associated these trends with an increase in the proportion of convective
storms during summer. Indeed, our model shows an increase in the proportion of storms that occur at higher temperatures (and therefore, following Eq. (3), have higher potential for extremes), which likely occurred during the summer (see Fig. 7 and 3). Moreover, our projections show an increase in the tail heaviness of $F(x)$, consistent with what was reported by Dallan et al. (2022). As the magnitude model is kept unchanged, this change in tail heaviness is mainly due to changes in the temperature distribution.

## 4 Case study

We next demonstrate how the TENAX model can be applied to a real-world climate change impact study. Our case study focuses on eight climate stations in Switzerland (Fig. 8). Located within or near the Swiss Alps, these stations represent various climates (Rubel et al., 2017): from warm temperate (Koppen classification Cfb) to boreal (Dfc) and Alpine (ET); and experience varying degrees of extreme precipitation (ranging from 6.3 mm to 25 mm for 10-minute extreme precipitation on





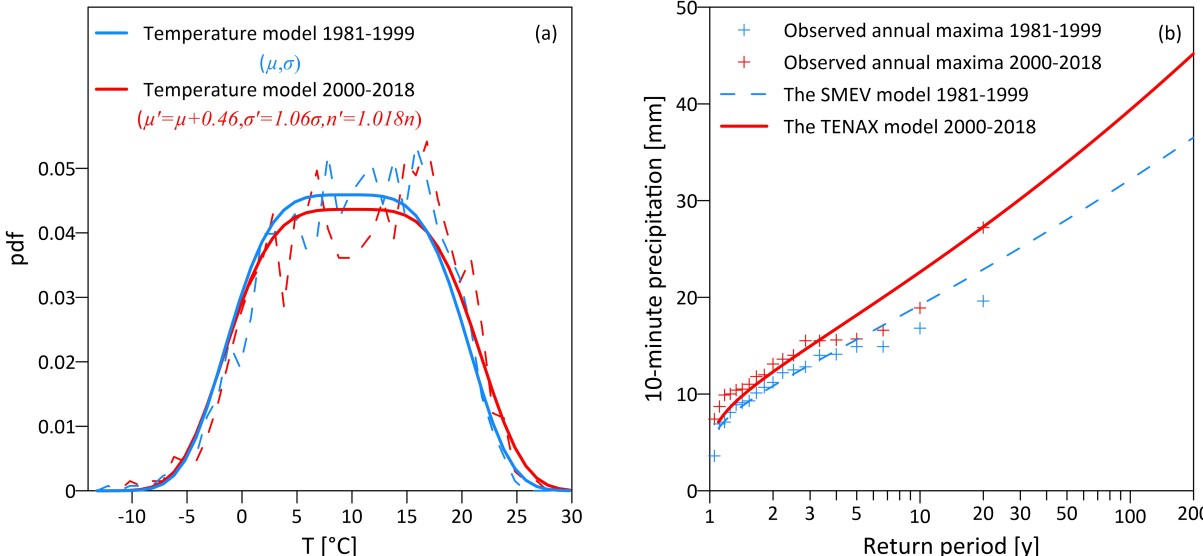

**Figure 7.** Predicting precipitation return levels for the period 2000-2018 for the Aadorf station based on temperature shifts with respect to the period 1981-1999. (a) The observed probability distribution function for the first (dashed blue line) and second (dashed red lines) periods and the temperature models fitted for the first period using the observed data (solid blue line) and projected for the second period by applying changes in $\mu$, $\sigma$, and $n$ to the model of the first period (solid red line). (b) The observed precipitation annual maxima for the first and second periods (blue and red plus symbols, respectively) and the estimated precipitation return period by the SMEV models fitted by the observed data for the first period (dashed line). The red solid line represents the precipitation return levels simulated by the TENAX model for the second period based on the magnitude model $W(x;T)$ fitted for the first period and the projected temperature model for the second period.

a 10-year return level; see Fig. 8). Furthermore, the climate stations are located at different elevations ranging from 273 m to 3294 m above sea level (Table S1), which makes them subject to varying degrees of orographic effects (Dallan et al., 2023) and projected temperature shifts (Palazzi et al., 2019). The complexity of the terrain and climate, and the availability of high-quality monitored and modeled data that are processed by a single operator (see next), provide an excellent basis for evaluating our model.

**4.1   Data**

A 10-minute precipitation and 1-hour temperature records were obtained for each station from MeteoSwiss for the period 1981-2018. The stations are part of the SwissMetNet project, which comprises about 260 automatic stations with strict quality control (Landl et al., 2009). We grouped the 10-minute precipitation intensities into precipitation events as described above (see also Marra et al. 2020) and calculated the average temperature during the 24 hours preceding the 10-minute peak intensity

of each event.

In addition, we obtained daily precipitation and temperature time series for each climate station based on the official CH2018 Swiss climate change scenarios (Sørland et al., 2020; Fischer et al., 2022). The daily time series are quantile-mapped and bias-





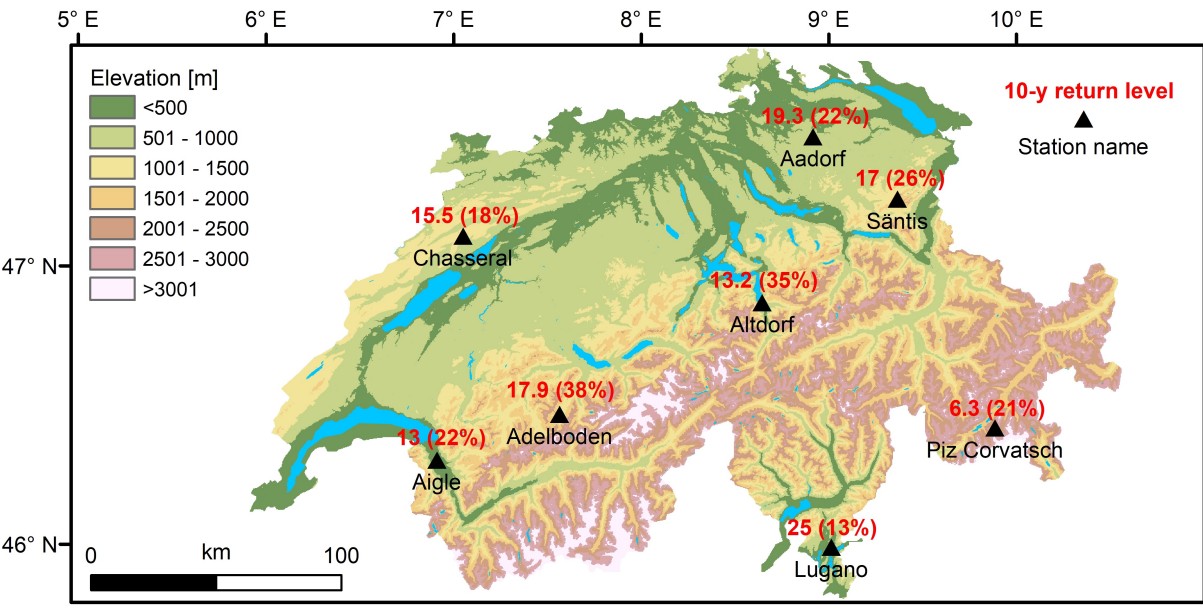

**Figure 8.** Location of the 8 sub-hourly climate stations. The values on top of each station indicate the 10-min peak precipitation [mm] for a 10-year return level as computed by MeteoSwiss. The values in brackets represent the intensification projected by the TENAX model for the end of the century (2081-2099) in comparison to the present climate (1981-2018) for the RCP8.5 emission scenario using a multi-model median projection from 10 climate models.

corrected for the period between 1980 and 2099, and are available for 10 regional climate models (see Table S2) following the CMIP5's RCP8.5 emission scenario. As a reference period (i.e., the current climate), we consider 1981 to 2020, while the future period for which we later compute changes in precipitation return levels is 2080 to 2099 (i.e., the end of the century). The changes in the annual occurrence of precipitation events $n$ were computed as the changes in the occurrence of daily rainfall events, and the projected changes in mean temperature $\mu$ and standard deviation of temperature $\sigma$ have been estimated based on the wet day temperatures. Table S2 presents the projected changes for each climate model, from which the multi-model median change for every station is calculated (Table 1). The changes in mean and standard deviation of temperature (which the model is sensitive to; Section 3.1) differ considerably from station to station (by up to $1.2°C$ and 12%). On the other hand, there is general agreement that the occurrence of annual precipitation events will decrease by approximately 4-7% for almost all stations toward the end of the century.

## 4.2 Present climate

Similar to the procedure described in Section 2 and demonstrated above for the Aadorf station, the parameters of the TENAX model were fitted to the observed precipitation and temperature data of the 8 stations (Table S3). Model results were validated against the official MeteoSwiss estimation of precipitation return levels (Table 2). There appears to be a good agreement



**Table 1.** Multi-model median changes in mean temperature $\mu$, temperature standard deviation $\sigma$, and the annual occurrence of precipitation events $n$ computed from 10 climate models (see details in Table S3). The changes are for the end of the century (2080-2099) in comparison with the reference period of 1981-2020, for the RCP8.5 emission scenario.

| Name | $\mu' = \mu+$ | $\sigma' = \sigma\cdot$ | $n' = n\cdot$ |
| --- | --- | --- | --- |
| Aadorf | 2.8°C | 0.99 | 0.93 |
| Adelboden | 3.1°C | 1.07 | 0.96 |
| Aigle | 2.3°C | 1.05 | 0.92 |
| Altdorf | 2.7°C | 1.10 | 0.99 |
| Chasseral | 3.0°C | 1.05 | 0.91 |
| Lugano | 2.5°C | 1.04 | 0.94 |
| Piz Corvatsch | 3.5°C | 1.11 | 0.95 |
| Säntis | 3.3°C | 1.03 | 0.95 |

between TENAX 's simulated precipitation return levels and MeteoSwiss's estimation, with an average bias of only 5.3% between the two estimations. The largest bias is found at Piz Corvatsch station for the 100-year precipitation return level (19.7%), but it remains within the uncertainty range of the estimation provided by MeteoSwiss (not shown). In fact, all of the TENAX model estimations fall within MeteoSwiss's uncertainty range. We note that while we fitted the model parameters for the period 1981-2018, MeteoSwiss' estimations are for a longer period (1960-2020).

### 4.3 Future climate

The TENAX model projections of precipitation return levels for the end of the century are presented in Fig. 9 for the Aadorf station; the estimations for the other stations are shown in Fig. S3. As expected, there is a high degree of heterogeneity in the estimations of the future precipitation return levels when examining the individual models (Fig. 9, gray lines). This is primarily due to the large uncertainties in the projections of change in temperature between the climate models (Fig. S2). Moreover, we see that the climate model uncertainty is more pronounced in long return levels, i.e., the model uncertainty increases in its prediction of precipitation return levels for the future climate (increased dispersion between gray lines in Fig. 9) as frequency decreases.

Precipitation return levels at the end of the century are increasing at all stations as a result of the projected increase in temperature at all locations (Fig. S3). However, the rate of intensification is not the same everywhere; as an example, precipitation is projected to increase at a rate ranging from 13% to 38% at the 10-y return level (Fig. 8). Intensification rates are not solely determined by temperature shifts and changes in precipitation occurrence, but also by the local physical processes behind 10-minute peak intensities (i.e., the parameters of the magnitude model). For example, while the changes in temperature and precipitation occurrence are rather similar between Lugano and Altdorf stations, the 10-year intensification in Lugano is projected to be in the order of 13% by the end of the century, while the intensification in Altdorf is anticipated to be in the order



**Table 2.** Official estimation of precipitation return levels (RL; years) made by MeteoSwiss for the period 1960-2020 (MS; mm) and the estimation of the TENAX model (mm) for eight stations examined in the case study for the period 1981-2018.

| | Aadorf | | Adelboden | | Aigle | | Altdorf | | Chasseral | | Lugano | | Piz | | Säntis | |
|---|---|---|---|---|---|---|---|---|---|---|---|---|---|---|---|---|
| RL | MS | TENAX | MS | TENAX | MS | TENAX | MS | TENAX | MS | TENAX | MS | TENAX | MS | TENAX | MS | TENAX |
| 2.33 | 12.7 | 12 | 11 | 10.9 | 8.3 | 8.6 | 8 | 8.4 | 9.7 | 9.6 | 16.7 | 16.7 | 4.1 | 4.2 | 11.3 | 10.8 |
| 5 | 16.4 | 16.1 | 14.6 | 14.9 | 10.7 | 11.3 | 10.7 | 11.6 | 12.8 | 12.6 | 21.1 | 21.3 | 5.2 | 5.2 | 14.3 | 13.4 |
| 10 | 19.3 | 19.8 | 17.9 | 18.5 | 13.0 | 13.7 | 13.2 | 14.4 | 15.5 | 15.2 | 25.0 | 25.3 | 6.3 | 6.1 | 17 | 15.7 |
| 20 | 22.4 | 23.7 | 21.5 | 22.3 | 15.6 | 16.2 | 15.9 | 17.5 | 18.3 | 17.8 | 29.2 | 29.2 | 7.5 | 6.9 | 19.8 | 17.9 |
| 30 | 24.1 | 26 | 23.7 | 24.6 | 17.3 | 17.6 | 17.7 | 19.3 | 19.9 | 19.4 | 31.8 | 31.5 | 8.3 | 7.5 | 21.5 | 19.3 |
| 50 | 26.5 | 29.1 | 26.8 | 27.7 | 19.8 | 19.6 | 20.1 | 21.8 | 22.2 | 21.4 | 35.2 | 34.4 | 9.5 | 8.1 | 23.7 | 20.9 |
| 100 | 30.1 | 33.4 | 31.6 | 32 | 23.7 | 22.3 | 23.9 | 25.3 | 25.5 | 24.3 | 40.6 | 38.5 | 11.2 | 9 | 27.1 | 23.2 |

of 35%. Furthermore, we observe that the rate of intensification tends to be negatively correlated with the precipitation return level. The intensification rate at Aadorf station, for instance, decreases from 22.5% to 19% when comparing 10- and 100-year return periods (Fig. 9).

## 5 Final remarks on the model and future development

We intend to introduce here the new TENAX model, explain the physical-based concept behind it, and demonstrate its abilities. There is, of course, a need to further develop the model and test it in other climates and environments in addition to the case study presented here. Considering the temperature model, future research should investigate the physical drivers underlying its form. For example, we demonstrated that in our case study a Generalized Gaussian model describes well the distribution of temperature in the 24 hours preceding sub-hourly peak intensities; this model emerges from the superposition of two seasonal Gaussian models (Fig. 3) with similar variance and almost equi-populated. It would be beneficial to test this temperature model in other locations as well as to propose general approaches to determine its form. We used a Weibull distribution for the magnitude model. In general, other locations could be described by different distributions, a possibility to be explored from case to case. It should be recalled, however, that the Weibull model adopted in our application yields parent distributions with different tails depending on the used temperature model (Fig. S2).

The TENAX model has the potential to be easily generalized to different types of precipitation (or seasons, or synoptic systems that initiate precipitation). In this case, each type of event would have specific magnitude and temperature models, and their impact on extreme return levels would be quantified using a multi-type SMEV (as in Marra et al., 2021). The combination of this approach with information on how temperature and precipitation occurrence are projected to change per event type may enhance the physical robustness of our projections.

We have briefly discussed the model's sensitivity and uncertainty issues. While a method to estimate the TENAX model's uncertainty in return levels for the present climate was presented and the sensitivity to the model parameters $\mu, \sigma$, and $n$ has been adequately explored (Section 3.1), we have merely touched on the uncertainties emerging from the TENAX model due to

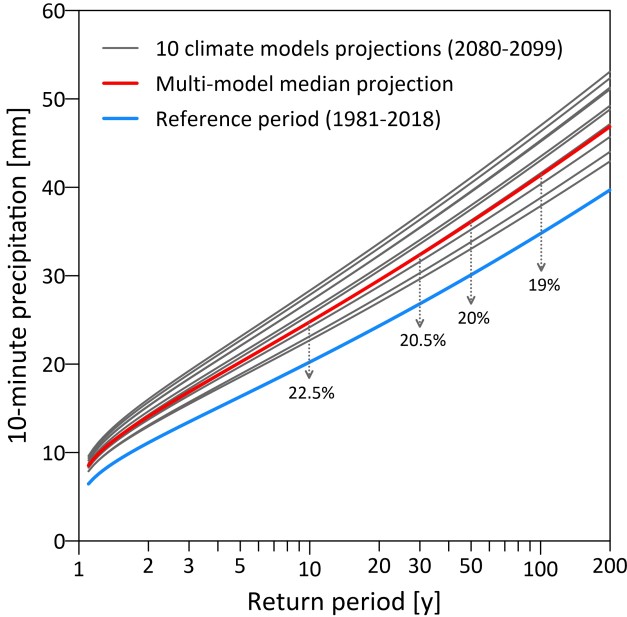

**Figure 9.** Extreme precipitation return levels computed by the TENAX model for the present (1981-2018; blue line) and future climate (2080-2099; grey lines represent 10 individual climate models and the red line shows the multi-model mean) for Aadorf station. Numbers represent the intensification rate for the 10-, 30-, 50-, and 100-y return periods.

climate model uncertainties in the projection of temperature and precipitation. Future developments will be necessary in this

regard.

Our final point is that it would be interesting and useful to develop further and apply the model to estimate the intensification of sub-hourly precipitation in urban areas, where the risk of pluvial flooding is high. The TENAX model will likely be required to be further developed to explicitly account for the urban heat island and its diurnal cycle, which is a critical physical component for the evaluation of extreme precipitation (Huang et al., 2022). Thus, a diurnal cycle module will likely need to

380 be incorporated into the model. These exciting directions highlight the great potential of using the TENAX model to predict changes in future sub-hourly extreme precipitation and related hazards.

## 6 Conclusions

We present the TENAX model, a physically consistent method to provide projections of extreme sub-hourly precipitation return levels in future climates based on the projected changes in temperature during precipitation events and in their occurrence

frequency. The TENAX model consists of two parts: a magnitude model, which represents the exceedance probability of sub-hourly intensity as a function of temperature, and a temperature model, which represents the probability density of temperatures during the events. The magnitude model contains information on the physics of the precipitation process and is assumed

invariant in time, which we demonstrate to be true using the model in hindcast. The temperature model changes in time and is used to estimate future return levels. There are only seven parameters in the model, which can be easily fitted to the data, making it simple and not demanding. The model can effectively be re-parameterized to fit future climate conditions by adjusting only three parameters, which can be estimated from a climate model at a daily resolution without the need to downscale precipitation or temperature data.

We showed that the TENAX model quantifies the observed return levels satisfactorily and with reduced uncertainties with respect to official estimates. It fully reproduces the observed scaling rates between extreme precipitation and temperature and can explain other known properties of this relation, such as the "hook structure", despite not being explicitly designed for it. The TENAX model represents a step forward in the understanding of the physical mechanisms behind the statistics of sub-hourly extreme precipitation and can be potentially used to assess changes in extreme sub-hourly precipitation more easily, accurately, and credibly than is currently possible.

*Code and data availability.* The TENAX model is available at https://doi.org/10.5281/zenodo.8345905 (Marra and Peleg, 2023), including the data necessary to reproduce the results for the Aadorf station as an example. The codes for the identification of ordinary events and for running the SMEV model are available at https://doi.org/10.5281/zenodo.3971558 (Marra, 2020). Precipitation and temperature data for all the stations in Switzerland shown in the case study were provided by MeteoSwiss and can be freely accessible from the IDAWEB at https://gate.meteoswiss.ch/idaweb.

*Author contributions.* Conceptualization: FM, NP; software development: FM, AC; data preparation: NP, MK; formal analyses: FM, MK, NP; funding acquisition: FM, NP, AC; paper writing – original draft: FM, NP; paper writing – review and editing: FM, MK, AC, NP.

*Competing interests.* At least one of the (co-)authors is a member of the editorial board of Hydrology and Earth System Sciences.

*Acknowledgements.* This study was supported by the Department of Geosciences of the University of Padova ("TENAX" project), by the CARIPARO Foundation through the Excellence Grant 2021 ("Resilience" project), and by the Swiss National Science Foundation (SNSF), Grant 194649 ("Rainfall and floods in future cities").



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
