# Peer review of "Predicting extreme sub-hourly precipitation intensification based on temperature shifts"

_Hydrology and Earth System Sciences, 2023_

## Author Response (AR1)

November 29th, 2023
Prof. Dr. Manuela Irene Brunner
Editor
Hydrology and Earth System Sciences

RE: Paper HESS-2023-226

Dear Prof. Dr. Manuela Irene Brunner,

We appreciate your handling of our manuscript. The revised manuscript entitled "*Predicting extreme sub-hourly precipitation intensification based on temperature shifts*" is enclosed, as well as supplemental materials and, as requested, a letter of response to all the remarks made by the reviewers. All the concerns raised by the reviewers have been addressed in the revised manuscript. We are confident that the manuscript can be considered for publication in *Hydrology and Earth System Sciences*.

Listed below are our responses (in blue) to the comments and suggestions of the reviewers. Line numbers refer to the "track changes" version of the manuscript.

Sincerely,

Nadav Peleg and Francesco Marra

On behalf of Marika Koukoula and Antonio Canale

**Reviewer #1**

This study presents a model for sub-hourly precipitation extremes that integrates the distribution of the temperature before the precipitation events. This model has the great advantage to have a relationship between extreme precipitation and temperature that is able to represent decreasing return levels for very high temperatures (above 25°C) which could have important implications for climate change impact studies. One important motivation for the TENAX model is that temperature is adequately represented by climate models, compared to precipitation (Fig. TS.2, IPCC, 2021). The fact that the TENAX models strongly rely on the distribution of temperatures before precipitation events is a great advantage in my opinion, and brings a lot of confidence in the resulting changes obtained from the model. Overall, the concept and the methodology are clearly presented and motivated. Based on my personal reading, this study is of interest for the research community working on extreme precipitation. However, there are several points that should be carefully addressed before publication and are listed below.

Thank you for your review. We are glad to see the strengths of our model were appreciated. Below you will find our response to the specific comments.

Validation of the approach

1. Throughout the manuscript, many conclusions are a bit oversold in my opinion. The use of the terms "validation" (l.200, l.201, l.213, l.267, l.268, l.330), "demonstrated" (l.289-209, l.356) are used and seems to indicate that the TENAX is the "true" model that should be used for any estimation of return levels of sub-hourly precipitation. However, any model has its assumptions and limitations, and the term "validation" has been criticized in this regard. Klemes (1986) proposed to speak about the operational adequacy of a model, rather than about its validity. Oreskes (1998) provides a detailed discussion on this subject and motivates the use of the term "evaluation" instead of "validation". I strongly advise the authors to replace "validation" with "evaluation" in the manuscript, because, in this study, the concept and the model are mostly illustrated rather than "validated". In particular, the title of the subsection "3.2 Validation of the projections" is particularly unfortunate because climate projections cannot be validated. They represent future scenarios based on a socio-economic pathway that will never occur. I recommend using "Hindcast model evaluation" instead.

Thank you for this suggestion. It is true that the term validation may be objected to, while the term evaluation may be preferred. Previously, the term "validation" was used as it is common practice, but we have amended the text to address this subject. The term "validation" or similar is replaced by "evaluation", and the title of 3.2 is modified to *"Hindcast evaluation of TENAX projections"*.

2. There are no quantitative metrics that demonstrate the better performance of the TENAX model over a recognized benchmark (if it exists). Different experiments are performed and are useful examples of how the TENAX model behaves (Figs. 4, 5, 7). However, at several times in the paper, other models considered as "best references" (MeteoSwiss estimates, the SMEV model in Fig. 4, "with reduced uncertainties with respect to official estimates" at l. 393-394) are used to show the superiority of the TENAX model whereas 1. These comparisons are mostly qualitative, 2. These "best references" also have their limitations/uncertainties.

Unfortunately, when it comes to return levels, no "true" reference is available to objectively benchmark one's model. The practical way we used to evaluate the performance of our model is to compare it with other available models in the literature. Here we selected one model based on standard asymptotic theory (MeteoSwiss, using the GEV model) and one alternative model based on a non-asymptotic approach (using the SMEV model). Both models are considered trustworthy and are commonly used by hydrologists and climatologists, and can therefore be used as a benchmark for our TENAX model.

One important aspect should be noted here. The aim of TENAX is not to replace or compete with these two models (or any model for stationary precipitation frequency analysis). Rather, it is to develop a model that allows projections based on temperature changes in a physically consistent manner. Consequently, a direct quantitative comparison with these models is not of interest. It is, however, important to ensure that the TENAX model can decently reproduce current climate conditions before moving forward with further analyses (i.e., climate change applications).

Convection-permitting simulations

3. In the manuscript, the use of simulations from convection-permitting models is discouraged at l. 49-52 and l. 239-240 on the basis that they are available for a few socio-economic pathways, and available with delay with respect to GCM simulations. These arguments are not fair in my opinion. First, concerning the emission scenario, the current manuscript only exploits simulations with the RCP8.5 and does not illustrate the advantage of having a multi-scenario ensemble. Second, regional climate simulations are also obtained with a lot of delay compared to GCM simulations. Concerning the CMIP5 experiment, for example, GCM simulations were made available in 2011 (Taylor et al., 2011) whereas some corresponding EUROCORDEX simulations have been made available in the past two years (e.g. with the RCM HadREM3-GA7). Furthermore, it is true that there are currently a limited number of convection-permitting simulations, but I would add that they are becoming increasingly available and that the next climate scenarios produced for Switzerland will make them available (CH2025, https://www.meteoswiss.admin.ch/about-us/research-and-cooperation/projects/2023/climate-ch2025.html).

We did not intend to discourage the use of CPM but rather to suggest an alternative for situations in which CPMs (or multiple CPM climate scenarios) are not available. There is no doubt that the availability of CPM in Switzerland, and across Europe in general, has increased significantly in recent years. In spite of this, CPM has not been applied to many other regions in the world. The text and tone have been revised to clarify this point.

The text that appeared in lines 49-52 of the original manuscript has been modified to read: "*Convection-permitting models can be used for such a task (Ban et al., 2020). However, these models are not available for all regions and are not forced by all the socio-economic pathways*". We have removed the text found in lines 239-240 of the original manuscript.

4. Besides these arguments, I find that the manuscript overlooks the limitations of regional climate simulations in terms of the reproduction of intense precipitation events. RCM simulations not only fail to simulate convective events, but they are also very limited concerning the moderate intensities (e.g. greater than 10 mm), the number of precipitation events, and their extent (Caillaud et al., 2021). These limitations have an important impact on the illustration of the proposed method since it exploits the changes in the number of precipitation events from the climate simulations (Fig. 6c). Even if the climate simulations have been corrected using quantile mapping methods, I am not confident that changes in terms of number of precipitations events can be properly obtained from regional climate simulations using parameterized convection.

We do agree with the reviewer that a note on the general limitations of regional climate models should be included. In line 395, we have added the following: «*While temperature simulations are regarded as relatively reliable, previous studies have indicated that regional climate models, such as the one we used, are unable to simulate intense convective precipitation events, often underestimating their frequency and number (e.g., Caillaud et al., 2021)*". We thank the referee for suggesting this addition to the text.

We would like to point out that our methodology eliminates the need for quantile mapping and bias correction of data from climate models, thus reducing the uncertainties in estimating how sub-hourly precipitation extremes will change as a result of global warming. Additionally, we demonstrate in Fig. 6 that changes in the sub-hourly return level are relatively insensitive to changes in the number of precipitation events.

Censoring threshold

5. A minor comment concerns the censoring threshold. I understood in Marra et al. (2019) that the motivation for this threshold was the estimation of the parameters of the distribution. A censoring threshold avoids the influence of very small and small intensities on the fitting. This approach is also proposed by Naveau et al. (2016) concerning the Extended GPD distribution. However, at l. 120-121, it seems that model 3 is only defined for values above the censoring threshold. I am also confused with the fact that this censoring threshold is defined in terms of possible range for the intensity x but at l.140, it seems to be a probability. I guess that it should be the quantile corresponding to this probability 0.9 but it needs to be clarified.

Thank you for pointing this out. We agree that our text was unclear on these lines. The threshold is used for left censoring. It follows that the model describes all the data (as the EGPD) and not only the exceedances. Differently from the EGPD, the model parameters are estimated to fit well only the values above the threshold and not all of them. The sentence was rephrased to (line 123): *"The Weibull tail model describes the non-exceedance probability of magnitudes using two parameters (scale and shape)..."*. And we describe the specific on the left censoring to the lines that follow the correction.

As the reviewer correctly points out, the values of the threshold are here expressed in percentiles (probability) and not in absolute terms. This is done in accordance with the practice of the SMEV method. To make this clearer, we rephrased the sentence in line 142 to *"Here we use a generic threshold $\vartheta^*$ equal to the local 90th percentile of the ordinary events"*.

Additional comments

6. l.96-98: Here, the authors refer to MEV distributions, is that correct? It could be indicated.

We rephrased to: *"These methods include the Metastatistical Extreme Value (MEV) and the Simplified MEV (SMEV), and allow one…"*

7. l.101: "Magnitude of the ordinary events". At this point of the text, it is not clear what these ordinary events refer to since they are defined later in section 2.1, and it is also unclear what the corresponding magnitude refers to (mean intensity over an event, maximum intensity).

Thank you for pointing this out. We rephrased line 102 to: *"...the cumulative distribution function of the magnitude of the events (see Section 2.1 for the definition of these magnitudes) occurring at…"*

8. l.112: The part "ordinary events of duration d are defined" is very confusing. I understand here that d is the duration of a storm whereas it seems to be a duration of interest as explained in Marra et al. (2020). I suggest providing an example or a schematic at this step, because the reader could easily get lost at the beginning of these technical explanations. This schematic could also illustrate the corresponding temperature T used in model 3 (see comment for l.160-161).

Following Marra et al. 2020, we use two concepts: the concept of "storm" and the concept of "ordinary event" (or "event"). Storms are independent meteorological objects, in our case the time series of precipitation intensities separated by dry periods of at least d_dry hours. Ordinary events are the quantities of interest of our analysis, in our case the peak intensities observed in each storm over a duration of interest. The concept of duration is the one typically used in statistical hydrology and should not be confused with the duration of the storm. As the reviewer mentioned, the above concepts are described in detail in Marra et al. (2020) and are used in many papers (e.g., Araujo et al., 2023; Dallan et al., 2022; Dallan et al., 2023; Formetta et al., 2022; Marra et al., 2021; Marra et al., 2022; Rinat et al., 2021; Shmilovitz et al., 2023). Therefore, we decided not to add any additional information or a schematic illustration to the manuscript describing the "event" concept to keep it as lightweight as possible, since it already contains a significant amount of information about the new models we introduce.

Araujo D, F Marra, H Ali, HJ Fowler, EI Nikolopoulos, 2023. Relation Between Storm Characteristics and Extreme Precipitation Statistics Over CONUS. Adv. Water Resour., 178, 104497, https://doi.org/10.1016/j.advwatres.2023.104497

Dallan E, M Borga, M Zaramella, F Marra, 2022. Enhanced summer convection explains observed trends in extreme subdaily precipitation in the Eastern Italian Alps. Geophys. Res. Lett., 49, e2021GL096727. https://doi.org/10.1029/2021GL096727

Dallan E, F Marra, G Fosser, M Marani, G Formetta, C Shäer, M Borga, 2023. How well does a convection-permitting regional climate model represent the reverse orographic effect of extreme precipitation? Hydrol. Earth Sys. Sci., 27, 1133-1149, https://doi.org/10.5194/hess-27-1133-2023

Formetta G, F Marra, E Dallan, M Zaramella, M Borga, 2022. Differential orographic impact on sub-hourly, hourly, and daily extreme precipitation. Adv. Water Resour., 149, 104085, https://doi.org/10.1016/j.advwatres.2021.104085

Marra F, M Armon, M Borga, E Morin, 2021. Orographic effect on extreme precipitation statistics peaks at hourly time scales. Geophys. Res. Lett., e2020GL091498, https://doi.org/10.1029/2020GL091498

Marra F, M Armon, E Morin, 2022. Coastal and orographic effects on extreme precipitation revealed by weather radar observations. Hydrol. Earth Syst. Sci., 26, 1439–1458, https://doi.org/10.5194/hess-26-1439-2022

Rinat Y, F Marra, M Armon, A Metzger, Y Levi, P Khain, E Vadislavsky, M Rosenaft, E Morin, 2021. Hydrometeorological analysis and forecasting of a 3-day flash-flood-triggering desert rainstorm. Nat. Hazards Earth. Syst. Sci., 21, 917–939, https://doi.org/10.5194/nhess-21-917-2021

Shmilovitz, Y., Marra, F., Enzel, Y., Morin, E., Armon, M., Matmon, A., et al. (2023). The impact of extreme rainstorms on escarpment morphology in arid areas: Insights from the central Negev desert. Journal of Geophysical Research: Earth Surface, 128, e2023JF007093. https://doi.org/10.1029/2023JF007093

9. l.123: Model 3 is defined for x>V*, does it mean that W(x;T)=0 for x<V*?

Please refer to the reply comment above. Briefly, the threshold is a left censoring threshold, so the model describes all the data (as the EGPD) and not only the exceedances. Differently from the EGPD, the model parameters are estimated to fit well only the values above a threshold and not all of them. This confusion has been avoided by rephrasing the text.

10. l.150: Please introduce the contents of Fig. 2 and provide an interpretation of the results.

Our purpose in presenting Fig. 2 is to demonstrate to the readers what the model components look like for the first time. The content of Fig. 2 is succinctly explained in the caption of the figure. The interpretation of the results is discussed in Section 2.5 (and elaborated on using Fig. 4 and 5). The text has been revised in line 152 to refer to it: "... *see the description of the station and data in Section 4 and the discussion of the model fit in Section 2.5 …*".

11. l.160-161: I am a bit confused by this comment. At l. 154-155, I understand that T is the average temperature observed during the D=24h preceding the peak intensity of an event which can occur at any time of the day. Daily temperatures from climate change projections are usually available at a daily scale for fixed daily intervals (e.g. from 12:00 day D to 12:00 day D+1). What is the temperature T taken from climate change projections in that case?

You are correct in your comment. For observed data, we can use the temperature that occurs exactly before the peak of the event, whereas in climate models, we are restricted to choosing either the temperature during the day when the peak of the event occurs or the temperature occurring on the day prior to the peak. Later in the manuscript, when we discuss how climate change is incorporated into the model, we explain that we use the daily temperature of the wet days. We have revised the text (line 163) to make it more accurate: "*We focus here on the case D=24 hours, as daily temperatures are easier to derive from climate model simulations (using the temperatures on wet days, as explained in Section 3) and are therefore preferred for climate change projections*".

12. l.167-172: Similar results are shown for Swiss stations in Evin et al. (2019) which applies a skew exponential power (SEP) distribution. The SEP distribution encompasses the Generalized Gaussian distribution and can take the skewness into account. Their Fig. 6 shows that depending on the season and the climate, the skewness and/or the kurtosis can be important and disqualify the application of Gaussian distributions, for some cases because of the skewness (for the station Jungfraujoch located at the elevation of 3580 m) and for other cases because of the flatness (for all stations during the months from March until August), as the authors obtained in the submitted manuscript.

Thank you for providing the reference to this article. In fact, we were not aware of it and agree that the SEP parametric distribution could be a viable alternative to the Generalized Gaussian we currently use. The use of this distribution will be considered in the next version of the TENAX model, and the following sentence has been added to the text (line 176): "*One potential direction, which has already been demonstrated for Swiss stations, is the use of the Skew Exponential Power (SEP) distribution, which is a flexible parametric distribution proposed by Evin et al. (2019)*".

13. l.204: "using an established non-asymptotic method": I understand that the term "established" intends to indicate that the return levels obtained with this method can be considered as a benchmark method. I would suggest

indicating the reference Marra et al. (2020) and a description of the method given at l. 207 and avoid the term "established" which tends to oversell the SMEV model in my opinion.

*Thank you for the suggestion. We rephrased the sentence to:* *"the return levels estimated using a non-asymptotic method (Marra et al., 2020; see below)..."*

14. l.236: "can be approximated by changes in the daily temperatures during precipitation events". I guess that these daily temperatures correspond to the days that include the peaks of the precipitation events. What happens if a precipitation event occurs on several consecutive days? Is it possible to be more explicit on the choice of the daily temperatures?

*We understand the referee's concern. Nevertheless, in the tested stations storms rarely last almost the entire day and do not last for several consecutive days (please see our definition of an event). Indeed, this could pose a problem in other climate regions, such as those affected by Monson, where continuous rainfall is common. However, even in such areas the scaling between sub-daily extreme rainfall and temperature still holds when using temperature on a daily basis (e.g. Ali and Mishra, 2018 and Ali et al., 2021). Accordingly, we infer that our approach based on daily temperature should still be scientifically sound in other climates.*

*Ali, H., & Mishra, V. (2018). Contributions of dynamic and thermodynamic scaling in subdaily precipitation extremes in India. Geophysical Research Letters, 45, 2352–2361. https://doi.org/10.1002/2018GL077065*

*Ali, H., Fowler, H. J., Lenderink, G., Lewis, E., & Pritchard, D. (2021). Consistent large-scale response of hourly extreme precipitation to temperature variation over land. Geophysical Research Letters, 48, e2020GL090317. https://doi.org/10.1029/2020GL090317*

15. Moreover, I understand that the term "changes" refers to absolute changes in mean temperatures over a future period and a reference period, as explained later in section 3.2. I do not understand why the daily temperature preceding the peak intensity can be approximated by the temperature during the precipitation event and these projected changes. At l. 236, it is indicated that "the advantage of using D=24 hours […] becomes now clear" but it does not become clearer at this point, it was already indicated at l.160-161. I completely understand the motivation for using daily temperatures because it is what is available in the majority of the regional climate simulations, but the manuscript does not show illustrations that it is also a reasonable choice compared to other durations.

*When we state "the advantage of using D=24 hours..." we are referring to exactly what the reviewer has mentioned in their comment - that most climate models operate on a daily basis and that the fact that TENAX operates on a daily basis makes its use straightforward for end users. We have indeed tested the model on temperatures ranging from one hour to 24 h before an event. Compared with using an average of temperatures over the last 24 h, we found that considering temperatures one hour prior to the event produced noisier results, but not necessarily better results. An important advantage of using 24 h is that it covers the full daily cycle and gets rid of possible differential shifts in temperature at different times of the day. This allows one to use daily temperature from the projections with more confidence. In light of this, plus the motivation of remaining at the same temporal resolution as most climate models, and given that it is most common to analyze the scaling relationship between sub-daily extreme rainfall intensities and temperatures using daily temperature data (e.g., Ali et al., 2018, 2021, among many others), we have decided to fix D to 24 h in our application. However, we note that this is a model parameter that can be modified by the users if they so choose. We added a sentence to explain the use of D equal to 24 h in the text (line 244):* "*We found that the 24-hour temperatures preceding the peak precipitation intensities can be approximated by changes in the daily temperatures during precipitation events (not shown). This is in accordance with previous studies exploring the relationship between sub-daily extreme rainfall intensities and temperature (Ali and Mishra, 2018; Ali et al., 2021b)*".

*Ali, H., & Mishra, V. (2018). Increase in subdaily precipitation extremes in India under 1.5 and 2.0 °C warming worlds. Geophysical Research Letters, 45, 6972–6982. https://doi.org/10.1029/2018GL078689*

*Ali, H., Fowler, H. J., Lenderink, G., Lewis, E., & Pritchard, D. (2021). Consistent large-scale response of hourly extreme precipitation to temperature variation over land. Geophysical Research Letters, 48, e2020GL090317. https://doi.org/10.1029/2020GL090317*

16. l.274-276: I did not understand what has been tested here. Likelihood ratio tests are usually applied to test different competing models for the same data. Here, model 3 seems to have been fitted on two periods. I do not understand how two models fitted with different data, with different parameters can be used to test the similarity of the model. If a Gaussian distribution is applied on temperature data for a past and a future period, assuming just a shift in the mean of the distribution, you would obtain a similar likelihood but two different distributions. Even if the test is valid, in my opinion, the invariance of the magnitude model cannot be demonstrated by comparing their properties on two periods of 20 years. Most stationary tests for precipitation need very long time series to be significant given the large inter-annual variability of precipitation (see Section 3.2 in Slater et al., 2021). However, these two lines are not necessary if this assumption is made explicit, and the paragraph at l.244-252 was sufficient from my point of view.

We thank the reviewer for raising this point which allows us to be more specific about the empirical validation of the proposed procedure. Likelihood ratio tests are very general tests amenable to many different applications. The general theory says that under a null hypothesis H0, said L(θ) the likelihood function computed for the parameter θ in the parameter space Θ, the log-likelihood ratio $-2ln\left(\frac{supH0L(\theta)}{supL(\theta)}\right)$ can be used as test statistics. Specifically, in the numerator, the likelihood is maximized under the constraints of the null hypothesis H0, while in the denominator, the likelihood function is maximized without constraints. In Section 3.2, the null hypothesis is that W(x;T) is the same in the two periods, or in other words that the parameters of W(x;T) are equal in the two periods. Under H0, the parameters are estimated from the original data basically ignoring the division into two periods. The unconstrained maximization, instead, fits a separate set of parameters for each of the two periods. The lack of evidence to reject the null hypothesis that the two periods share the same set of parameters supports our conclusion that a magnitude model fitted for an observed period will be invariant in the future. The revised manuscript (Section 3.2) has been revised to include a more comprehensive explanation.

17. l.316-319: This part lacks important information and should be improved. First, it is indicated that the projections are obtained from 10 regional climate models, whereas they are obtained from 4 different RCMs (SMHI-RCA4,MPI-CSC-REMO2009,CLMcom-CCLM4-8-17, DMI-HIRHAM5) which have been to downscale 4 GCMs (ICHEC-EC-EARTH, IPSL-IPSL-CM5A-MR, MOHC-HadGEM2-ES, MPI-M-MPI-ESM-LR). Table S2 presents these 10 GCM/RCM combinations as different climate models but the RCMs and the GCMs have their own properties and limitations (some GCMs warm more than others due to their climate sensitivity). I understand that the space is limited in Table S2 but I suggest providing a separate table with the complete information of these simulations (GCM, RCM, GCM member). For example, the second line simply indicates "IPSL" which is the name of a French institution that produces many different climate simulations (from GCMs and RCMs) and not the name of a climate model. What is also missing is the spatial resolution of the climate simulations. The CMIP5-EUROCORDEX simulations have been produced at a 12 km resolution. In https://www.nccs.admin.ch/dam/nccs/de/dokumente/website/klima/CH2018_Technical_Report-compressed.pdf.download.pdf/CH2018_Technical_Report-compressed.pdf, it is indicated that quantile mapping is applied to station observations as well as gridded observations at 2 km to derive localized climate projections. Have station observations been used to provide the corrected climate simulations used in this manuscript?

We elaborated on the climate models that were used, as suggested. We have added information to the text to clarify that we used the 12-km resolution climate models, downscaled and bias-corrected based on the observed data, as the reviewer assumed. The text has also been revised to explicitly state that the 10 climate datasets are a combination of four different RCMs driven by four different GCMs. In addition, we have added a new Table S3 summarizing the GCM-RCM chains and their initial conditions used in this study.

18. l.326-327: I suggest replacing "occurrence of annual precipitation events" with "average number of precipitation events at an annual scale". In table S2, where does the decrease of 4-7% come from? The minimum n. values for the different stations are 0.84, 0.82, 0.78, 0.81, 0.83, 0.74, 0.82.

The values refer to the median change at the annual scale presented in Table 1. We revised the text as suggested and added a reference to Table 1.

19. Table 1, Figure 9: I do not recommend using median/mean statistics obtained from a multi-model ensemble. The climate models have different biases and limitations and are not independent (Brunner et al., 2020). The mean or median of this ensemble cannot be considered as a "best estimate". While the use of multi-model ensembles is now the norm in climate change impact studies, the "one model / one vote" approach has been criticized in many studies (Tebaldi and Knutti, 2017; Abramowitz et al., 2019), as well as considering a multi-model ensemble as a "sample" (von Storch and Zwiers, 2013). I strongly recommend providing intervals of the changes in Table 1 and removing the red line in Figure 9.

*First, we completely agree with the reviewer's comment and in general advise against using multi-model means or medians, and instead present an ensemble of changes. We note, however, that the purpose of our paper is not to discuss and present the magnitude of intensification of extreme rainfall associated with increasing temperatures and associated uncertainties in Switzerland, but rather to illustrate how the TENAX model can be employed to quantify this change. Therefore, we simplified the discussion by focusing on the median change. Nevertheless, we do present the ensemble of changes for all climate models used in the analysis of all climate stations in Fig. 9 and Fig. S3, which readers may use as an indication of the uncertainty of the predictions. We also illustrate the temperature change uncertainty resulting from the climate model in Fig. S2. We added the following sentence in Section 4.3 (line 367) to clarify the simplicity we have taken here: "... values represent median changes. The ensemble in Fig. 9 and S3 provide insight into the uncertainties arising from different climate models …".*

20. l.360-361: "this model emerges from the superposition of two seasonal Gaussian models": this might be the case in your example but that is not necessarily true, i.e. generalized Gaussian models with heavy tails can also be obtained at a monthly scale (Evin et al., 2019).

*We agree. Please note that this is a continuation of the previous sentence where it is explicitly implied that we refer to our specific case study.*

21. l.477: I did not find a reference "Marra et al. 2021b" and only "Marra et al. 2021" is cited at l.60. Conversely, there are two references Marra et al. (2023).

*Thank you for pointing this out. The reference is updated to Marra et al. 2021. Conversely, the 2023 references are correct, as one is Marra et al. 2023, and the other is Marra and Peleg 2023.*

**Reviewer #2**

General comments:

The authors provide a physical based statistical approach to estimate future sub-hourly extreme rainfall. The main idea is using an event based non-stationary Metastatistical Extreme Value (MEV) distribution for rainfall, the Generalized Gaussian distribution for the conditioning temperature of the events and accounting separately for the frequency of events. The change in temperature and occurrence of events is provided by climate models. They validate the approach on a hindcast experiment, assess uncertainties and finally apply the framework in a case study to project 10-min extreme rainfall for 8 climate stations in Switzerland. The idea is novel and significant, clearly explained as well as objectively validated. The manuscript is short, well written to the point avoiding unnecessary text burden. The developed software is freely provided. From my point of view, this is an excellent paper and I have only minor suggestions for improvement (see below).

*Thank you for taking the time to review our manuscript. We are glad to see the strengths of our model were appreciated. Please see below our response to the specific comments.*

Minor specific comments:

1. Line 80: "Using current methods it is thus impossible …" I would not be so strong and recommend to replace "impossible" by "highly uncertain" or a similar term.

*The text has been revised as suggested.*

2. Section 2.1/ 4: It is known, that the performance of the MEV approach depends strongly on the correct selection of the underlaying probability distribution for the ordinary events. Here the Weibull distribution is selected without much discussion. I would propose at least to provide goodness of fit test results for the case study.

Please refer to our reply to the next comment.

3. Section 2.2/ 4: Similarly, I also would propose to show goodness of fit test results for the applied Generalized Gaussian distribution for the temperature data.

We reply here to both comments 2 and 3.

The reviewer is correct: the performance of the non-asymptotic approaches strongly depends on the underlying probability distribution of the ordinary events. We, however, disagree on the use of goodness of fit metrics for these purposes. Goodness of fit is not always the optimal way to determine whether a statistical model is appropriate. In particular, this is true for extremes, which represent small and highly noisy samples. The robustness of our approach lies in the fact that we choose simple models with an underlying physical basis, and we show that they can (a) well reproduce extremes; (b) explain other phenomena not specifically designed to be explained by the model (Fig. 5 and the discussion that follows); and (c) explain tail behaviors different from Weibull that were reported by other authors (e.g., Berg et al., 2013).

The Weibull distribution is not used here to model the ordinary events distribution (*F* in the manuscript), rather a non-stationary Weibull is used to model the conditional distribution of the ordinary events given the temperature (*W* in the manuscript). The (marginal) ordinary events distribution *F* emerges from the combination of this model with the distribution of temperature (*g* in the manuscript) and is a straightforward application of Bayes theorem. Note that, as shown in Fig. S2, our *F* model can produce ordinary event distributions that are not Weibull. For the temperature model, we would like to add that the shape parameter of the temperature model is explicitly fixed to 4 (or 2 in some examples such as Fig. 3 and S1) exactly to avoid overfitting caused by the optimization of some goodness of fit metric (in our case, it would be the log-likelihood - but any other criterion would have the same outcome).

We would have obtained better fits (a) using a free-shape parameter in the temperature model; (b) by focusing on the right tail of the temperature model (e.g., with a left-censoring); and (c) by exploring dependencies of our scale parameter on temperature that are not exponential. In all these cases, however, we would have disconnected our model from its possible physical interpretation. Our view is that physical interpretability should take precedence over goodness of fit.

Berg, P., Moseley, C., and Haerter, J.O.: Strong increase in convective precipitation in response to higher temperatures. Nat. Geosci. 6 (3), 181–185. https://doi.org/10.1038/ngeo1731, 2013

4. All text: The authors use "Montecarlo" in the text. This is quite uncommon. I would propose to write this term as "Monte-Carlo (MC)".

Thank you for the suggestion. We replaced all the occurrences in the paper with "Monte Carlo", which appears to be the most accepted terminology.

5. Line 282: … "the fitted SEMV models to both periods (dashed lines in Fig. 7b)". As I understood you fitted the SMEV to both periods. Accordingly, there should be two dashed lines in this figure. I can see only one dashed blue line? Is the line for the future period beneath the red line and or did you forget to put it in the figure?

The reviewer is correct - the dashed line for the 2000-2018 period overlaps with the solid TENAX line, so it has been removed. The manuscript and figure caption have been revised to refer to a single dashed line (SMEV, for the first period) and a single solid line (TENAX, for the second period).

Again, we would like to thank the Editor and the two anonymous reviewers for their comments and suggestions.

Sincerely,

Nadav Peleg and Francesco Marra

On behalf of Marika Koukoula and Antonio Canale